# Exopolysaccharide Production from Marine-Derived *Brevundimonas huaxiensis* Obtained from Estremadura Spur Pockmarks Sediments Revealing Potential for Circular Economy

**DOI:** 10.3390/md21070419

**Published:** 2023-07-23

**Authors:** Marta Catalão, Mafalda Fernandes, Lorena Galdon, Clara F. Rodrigues, Rita G. Sobral, Susana P. Gaudêncio, Cristiana A. V. Torres

**Affiliations:** 1Associate Laboratory i4HB, Institute for Health and Bioeconomy, NOVA School of Science and Technology, NOVA University of Lisbon, 2819-516 Almada, Portugal; 2UCIBIO—Applied Molecular Biosciences Unit, Chemistry and Life Sciences Departments, NOVA School of Science and Technology, NOVA University of Lisbon, 2819-516 Almada, Portugal; 3CESAM—Centre for Environmental and Marine Studies, Department of Biology, University of Aveiro, 3810-193 Aveiro, Portugal

**Keywords:** exopolysaccharides (EPS), biopolymers, marine-derived bacteria, blue biotechnology, circular economy, saline media

## Abstract

Marine environments represent an enormous biodiversity reservoir due to their numerous different habitats, being abundant in microorganisms capable of producing biomolecules, namely exopolysaccharides (EPS), with unique physical characteristics and applications in a broad range of industrial sectors. From a total of 67 marine-derived bacteria obtained from marine sediments collected at depths of 200 to 350 m from the Estremadura Spur pockmarks field, off the coast of Continental Portugal, the *Brevundimonas huaxiensis* strain SPUR-41 was selected to be cultivated in a bioreactor with saline culture media and glucose as a carbon source. The bacterium exhibited the capacity to produce 1.83 g/L of EPS under saline conditions. SPUR-41 EPS was a heteropolysaccharide composed of mannose (62.55% mol), glucose (9.19% mol), rhamnose (19.41% mol), glucuronic acid (4.43% mol), galactose (2.53% mol), and galacturonic acid (1.89% mol). Moreover, SPUR-41 EPS also revealed acyl groups in its composition, namely acetyl, succinyl, and pyruvyl. This study revealed the importance of research on marine environments for the discovery of bacteria that produce new value-added biopolymers for pharmaceutical and other biotechnological applications, enabling us to potentially address saline effluent pollution via a sustainable circular economy.

## 1. Introduction

The huge need to minimize dependence on fossil-fuel resources and the generation of solid and liquid wastes has led to the necessity for change toward a circular economy. With the increasing world population, industrialization, and urbanization, saline effluents from various industries, such as concentrated brine from various sources, such as desalination plants, oil and gas fields, mining, and other industries (e.g., textile, aquaculture, among others), result in an enormous number of pollutants with undesirable effects on human health and the environment, including threats to coastal ecosystems, as well as surface waters, damaging the marine environment due to the generation of anoxic conditions on the sea floor, changing light conditions, and influencing of the proliferation of marine species. The proper disposal and treatment of these effluents are persistent problems that pose significant technical and economic challenges. Regulatory pressure, public awareness and environmental issues have revealed the need for proper treatment of saline waste using innovative and cost-effective techniques, increasing interest in sustainable strategies for treating saline effluents, including the removal of organic pollutants from these effluents. Anaerobic/aerobic biological treatments feasibly remove carbonaceous, nitrogenous, and phosphorous pollution at high salt concentrations. The transformation of saline waste streams into valuable products, such as exopolysaccharides (EPS), can be enhanced by studying bacteria of marine origin, which use saline media for their growth [1].

Bacteria have a key role in this transition and are crucial to an effective circular economy, being capable of converting carbon sources (e.g., biological wastes) into value-added products (e.g., biopolymers, biogas, building block chemicals) [2]. Therefore, it is of upmost importance to discover new bacteria with innovative abilities and pathways to meet the needs of modern society.

The world’s oceans are still overlooked and rather unexploited, remaining an open and very promising resource for the discovery of new molecules [3,4]. More specifically, microorganisms are spread across the oceans and well adapted to adverse conditions (e.g., extreme pH, low temperature, salinity, osmotic pressure) [5]. Thereafter, bacteria developed the ability to produce unique biomolecules to survive in this type of habitat. Among these molecules are EPS, which are biopolymers excreted by several living beings, including microorganisms, plants, and algae. Nevertheless, molecules extracted from bacteria have improved physical and chemical properties compared to those extracted from other organisms [6]. EPS are produced in response to biotic or abiotic stress factors as extra biological protection for the cells [4]. These molecules can be found attached to the cell wall or surrounding the cells as a slime, which is weakly connected to the cell surface and contributes to biofilm formation [7,8,9]. EPS biopolymers are high-molecular weight carbohydrates, of which the most common sugar residues are glucose and galactose, although certain bacteria are capable to produce monosaccharides that are uncommonly found in nature, such as fucose, rhamnose, or uronic acid. Many EPS possess non-sugar substituents called organic acyl groups (acetyl, succinyl, pyruvyl) and inorganic groups (sulfate, phosphate) [10]. Over the last few years, the interest in EPS has increased in various industrial sectors, such as food, textiles, pharmaceuticals, medical, and cosmetics, as their environmentally friendly, novel, and unique physical and chemical characteristics enable diverse functional properties, such as acting as stabilizing, thickening, gelling, antioxidant, and emulsifying agents or scaffolds for tissue engineering and drug delivery [11,12,13].

Numerous microbial EPS structures have been reported, but only a few, such as xanthan gum, levan, gellan, and dextran, are well known industrial EPS with considerable markets [14]. However, in recent years, the demand for EPS produced by bacteria, specifically marine-derived bacteria, has increased, as terrestrial and marine environmental conditions are distinct, and bacteria adapt and evolve differently to survive. To date, few genera of EPS-producing bacteria isolated from extreme marine environments have been commercialized. The first commercialized marine-derived bacterial EPS was deepsane, produced by *Alteromonas macleodii*, having high molecular weight and composed of seven different sugar residues: fucose, rhamnose, glucose, galactose, mannose, glucuronic acid, and galacturonic acid [15]. This EPS is commercially available under the name of Abyssine^®^ and has widespread importance for the cosmetics industry because it smooths and reduces irritation of sensitive skin due to chemical, mechanical, and UVB exposure [16]. Another marine-derived bacterial EPS is HE800EPS (Hyalurift^®^, trademark), which is like hyaluronic acid, having a rare glycosaminoglycan-like structure, and is produced by *Vibrio diabolicus*, a deep-sea bacterium isolated from a Pompei worm tube (polychaete *Alvinella pompejana*), collected from a deep-sea hydrothermal field in the East Pacific Rise [8,17]. HE800EPS has great in vitro tissue regeneration potential, is capable of restoring bone and skin, and has the ability to accelerate in vitro collagen fibrillation and activate fibroblasts [18].

In this study, halophilic bacteria isolated from marine sediments collected from the Estremadura Spur pockmarks field (Portugal), at depths of 200–350 m were screened for the ability to produce rare EPS under saline conditions to study their effects regarding EPS yield and monosaccharide composition. The biosynthesized biopolymers were characterized in terms of composition. Pockmarks are a cold seep extreme environment.

## 2. Results and Discussion

### 2.1. Characterization of EPS Producing Bacterial Strain

A total of 67 bacterial strains were able to synthetize EPS. The higher yield producers, in 30 mL of M1 medium, revealed production between 0.20 g/L and 0.41 g/L. Of these strains, the 25 bacterial strains that revealed the highest EPS production were tested in a higher scale (200 mL medium M1), yielding higher growth rates (0.05–0.12 h^−1^) and achieving CDW values between 0.16 ± 0.02 g/L and 2.70 ± 0.20 g/L (Figure 1), corresponding to the strains SPUR-1 and SPUR-11, respectively. The general increase in cellular growth may have been a direct consequence of the improvement in essential parameters, namely stirring and oxygen availability (enabled by the higher head space). Concerning EPS production, the results obtained with the scale increase did not always result in higher EPS production. An example is SPUR-11, which showed production of 0.20 ± 0.01 g/L in 30 mL, but on a 200-mL scale showed slightly lower EPS production of 0.13 ± 0.09 g/L. However, the attained cellular growth was much higher (2.70 ± 0.20 g/L), suggesting a metabolic preference to grow using the available nitrogen and carbon sources, instead of secreting EPS. The strains SPUR-41, -47, and -55 reported the highest production of EPS (0.88 ± 0.02, 0.30 ± 0.07, and 0.30 ± 0.07 g/L, respectively) (Figure 1). It is worth highlighting that for the strains SPUR-28, -41, and -47, the EPS production yield was higher than their cellular growth. Even for cellular growth, the improvement in parameters such as stirring, and oxygen availability may have contributed to the improvement in EPS production. Although the pH was not controlled during the runs, its value practically did not change (pH between 6 and 7), which may have impacted the process positively for higher EPS production. According to the literature, most microorganisms have a preference to produce EPS in media buffered at a neutral and constant pH [19,20].

The obtained results were promising since the screening assays were performed without controlling the essential parameters for bacterial growth. Comparing our results with the literature [21], which studied the production of EPS from *Bacillus pseudomycoides* U10 and obtained EPS production of 0.09 g/L with a controlled pH of 7 and production of 0.11 g/L when testing the effect of a temperature of 37 °C, we found lower yield results than those obtained for several SPUR strains. Joulak and co-workers screened different *Halomonas* strains isolated from different Tunisian hypersaline environments for EPS production, achieving EPS concentrations between 0.09 and 0.17 g/L in media M1 and M2 [22]. Nevertheless, Roca et al., in similar growth conditions, studied the EPS production of bacteria isolated from the Madeira Archipelago ocean sediments, achieving concentration higher than those obtained in this study (0.99–6.88 g/L) [23]. Further, EPS production between 4.2 and 8 g/L was reported from Egyptian marine bacteria cultivated in a 250 mL shake flask with marine broth and glycerol [24].

The sugar residue composition of the EPS synthesized by the SPUR bacteria grown in 200 mL of medium M1 is presented in Figure 2. For all the evaluated EPS, the principal sugar constituents were neutral monomers, such as glucose and mannose. Glucose was present in all EPS from all strains in values between 15.43% mol and 47.54% mol. The sugar residue existing in the highest concentration was mannose, in a range from 4.70% mol to 73.68% mol. A deoxy-hexose also commonly produced by marine bacteria is rhamnose [25,26]. However, this monomer was only detected in EPS synthesized by bacteria SPUR-39, SPUR-34, and SPUR-69 with 14% mol, 2.96% mol and 29.5% mol, respectively. In contrast to rhamnose, fucose, a rare sugar, was produced by almost all SPUR bacteria, and SPUR-56 showed in its composition 31.10% mol, which is a very remarkable concentration. Although fucose is difficult to obtain in nature [3], it may confer to polymers biological activity, biofilm-forming properties, hydrophilicity, high permeability to water vapor, and good barrier properties to gases (CO_2_ and O_2_) [27]. For instance, FucoPol is a fucose-rich EPS that exhibits interesting functional properties, such as forming viscous aqueous solutions with shear thinning behavior, film-forming properties, flocculation activity, emulsion forming, and stabilizing capacities [6,27,28]. Another bacterial fucose-containing EPS is Fucogel, produced by *Klebsiella pneumonia* I-1507, commercialized by the cosmetics industry because of its thickening, emulsifying, and film-forming properties [14]. There have been reports of marine EPS that are heteropolysaccharides, of which mannose and glucose are the dominant sugars. Other authors have reported marine EPS containing fucose in their composition. One example was reported by Bramhachari et al. from a marine bacterium, *Vibrio harveyi* strain VB23, which produced an EPS with emulsifying properties and having fucose in its composition [29]. Comparing the EPS composition of the studied SPUR bacteria with *Pseudomonas* sp. ID1 and *Pseudoalteromonas* CAM025, both isolated from marine sediments collected in the Antarctic Sea, strains such as SPUR-8, -11, -28, and -64 produced fucose containing EPS in greater amounts (%) [30,31].

In most of the EPS produced by SPUR strains, arabinose was also detected in amounts between 0.63% mol and 6.28% mol. The presence of glucuronic acid has been described in some marine bacteria studies [23,32]. Caruso and co-workers reported a bacterium, *Marinobacter* sp., W1-16, from Antarctic surface seawater, which produced an EPS composed of glucose, mannose, galactose, galacturonic acid, and glucuronic acid with emulsifying, cryoprotective, and heavy metal-binding properties [33]. Uronic sugars, such as glucuronic and galacturonic acids, were present in the EPS of all strains, and SPUR-55 reported a yield of 17% mol of these rare acid sugars. These types of sugars are important for the cosmetic and medical industries due to their bone- and skin-restoring properties [34].

### 2.2. Taxonomic Identification of the Higher Yield Marine-Derived Bacteria EPS Producers

The three strains that exhibited the higher EPS yield, namely SPUR-55, SPUR-41, and SPUR-64, were taxonomically characterized based on the 16S rRNA gene sequencing data analysis. The producing strains are *Brevundimonas huaxiensis* (SPUR-41), *Bacillus* sp. (SPUR-55), and *Bacillus* sp. (SPUR-64) (Figure 3).

Effectively, the analysis of the 16S rRNA gene sequence of strain SPUR-41 revealed 100% sequence identity with the newly described *B. huaxiensis* [35] and 99% sequence identity with *B. vesicularis,* and *B. nasdae*. There have been reports of EPS *Brevundimonas* producers isolated from saline environments [36,37,38]. Although EPS production was previously described for a *B. vesicularis* strain isolated from a paper mill [39] and for *B. nasdae* isolated from the wastewater treatment site soil of a gold and copper mine [40], which secreted EPS under arsenite stress, there have been no reports of EPS production by a marine counterpart. Thus, our study is the first report of EPS production by a marine-derived strain of the *Brevundimonas* genus.

SPUR-55 revealed 100% identity with four different strains of the *Bacillus* genera (Figure 3). *Bacillus aerius* ATHM35, a salt-tolerant EPS producer, was isolated from hypersaline soils [41], and the deep sea psychrotolerant *B. altitudinis* SORB11 produced EPS with a glucomannan-like configuration, composed of → 4)-β-Man*p*-(1 → and → 4)-β-Glc*p*-(1 → residues) [42]. In another study, *B. altitudinis* MSH2014, isolated from mangrove trees in Ras Mohamed, Red Sea Coast, Sinai Peninsula, Egypt, produced EPS containing mannouronic acid, glucose, and sulfate, [38]. *B. aerophilus* rk1 isolated from the soil of a Nizam sugar factory, Nizamabad district, Telangana, India, also revealed EPS production with antioxidant activity [43]. *B. stratosphericus*, isolated from the deep Southern Ocean (Indian Sector), also revealed EPS production [44].

**Figure 3 marinedrugs-21-00419-f003:**
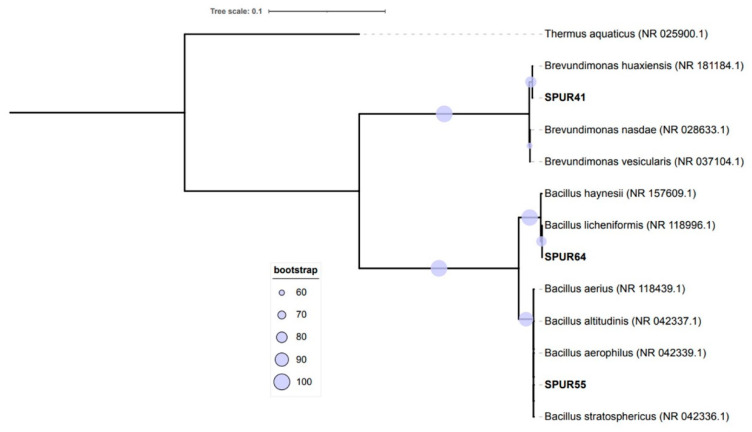
Phylogenetic tree of the alignments of the 16S rRNA gene for three different strains isolated from the Estremadura Spur pockmarks field (Portugal), namely SPUR-41, SPUR-55, and SPUR-64, using the evolutionary model GTR + I + G using jmodeltest-2.1.10 analysis (Vigo University, Vigo, Spain) [45]. Sequences were aligned using MAFFT v7.490 (Osaka University, Suita, Osaka, Japan) [46] and trimmed to 1300 bp with trimAL trimAl 1.2rev59 (Centre for Genomic Regulation, Barcelona, Spain) [47]. The tree was created using Iqtree v2.0.7 (University of Vienna, Vienna, Austria) [48] with 5000 bootstraps.

The strain SPUR-64 presented 99.9% identity with *Bacillus licheniformis* and 99.8% identity with *B. haynesii* (Figure 3). Spano et al. described *B. licheniformis* DSM 13 as a haloalkaliphilic and thermophilic bacteria, and *B. licheniformis* T14, also isolated from the marine environment, namely from a shallow hydrothermal vent of Panarea Island (Italy), was reported to be an EPS producer [49]. Strain T14, which is highly related to both DSM 13 and SPUR-64, produced an EPS containing fructose, fucose, glucose, glucosamine, and mannose. In contrast, the composition of EPS from SPUR-64 was fucose, arabinose, galactose, glucose, mannose, glucuronic acid, and rhamnose (Figure 2). *Bacilllus haynesii*, isolated from Campanario hot spring in the Central Andean Mountains of Chile, has also been reported to be an EPS producer [50].

### 2.3. Effect of NSW and Temperature on Bacterial Growth and EPS Production

Considering the cellular growth and the composition and production yield of EPS in the previous assays, SPUR-41, -55, and -64 were selected and grown in NSW at 30 °C and 19 °C (Figure 4) to evaluate their impact on biopolymer production.

SPUR-55 showed increased growth in NSW, from 1.03 ± 0.12 g/L to 1.4 ± 0.05 g/L. However, the greatest increase was obtained for SPUR-41, with a CDW of 1.87 ± 0.07 g/L in NSW, more than twice the CDW achieved in SSW (0.83 ± 0.01 g/L). These differences could be due to the presence of micronutrients that are essential for the growth of this bacterium and that are not present in SSW. The exception was SPUR-64, which reached a lower CDW value (0.84 ± 0.15 g/L) compared with the CDW achieved in SSW (1.02 ± 0.04 g/L).

Regarding EPS production, the same strains, SPUR-41 and SPUR-55, grown in NSW, showed increased production values. For SPUR-55, the EPS production increased from 0.3 ± 0.07 g/L when grown in SSW to 0.55 ± 0.06 g/L when grown in NSW. SPUR-41, which remained the strain with the highest production yield of EPS, produced 1.2 ± 0.02 g/L of EPS when grown in NSW. The only strain with a different profile was SPUR-64, which showed similar values of EPS production in both water types. These results indicate that the salt concentration in NSW, 1.0% (*w*/*v*), was more favorable for EPS production by SPUR-55 and -41 than the concentration of 2.3% (*w*/*v*) from the SSW. A study that reported the effect of NaCl, (0–20% *w*/*v*) on the production of EPS by a marine microorganism, *Hahella chejuensis* strain 96CJ10356, also concluded that the ideal concentration was 1% (*w/v*), while the increase in NaCl concentration led to a decrease in EPS production [51,52]. In contrast, SPUR-64 showed higher salt tolerance and produced EPS in a higher range of salt concentrations.

To evaluate the effect of temperature on EPS production, experiments were performed at 19 °C. Incubation at lower temperatures was previously described to result in the enhancement of EPS production and reduction in the growth rate and cellular mass, with a longer exponential phase [53]. In the mechanism proposed by Sutherland to explain this phenomenon, a decrease in temperature would cause a decrease in the growth rate and thus in the synthesis of cell wall polymers, which would in turn result in more precursors available for EPS production [10]. Although SPUR-41 and -55 showed decreased growth at 19 °C (0.85 ± 0.01 and 0.97 ± 0.13 g/L, respectively), with lower specific growth rates (0.04 and 0.06 h^−1^), the production of EPS was also lower than at 30 °C (0.73 ± 0.04, 0.18 ± 0.05 g/L, respectively). This behavior may be explained by a direct association between growth and EPS production or by the temperature for optimal EPS production being higher for these bacteria despite being isolated from a cold environment. In fact, the optimal temperature described for *Brevundimonas* to grow and produce EPS is 37 °C [35]. In the case of SPUR-64, the results showed a similar EPS production level when grown at 19 °C and 30 °C. Also, *Bacillus licheniformis* was previously described to have an optimal EPS production temperature of 37 °C [49], suggesting that, in the case of SPUR-64, higher temperatures may improve EPS production.

The sugar monomer composition of the EPS synthesized by these SPUR strains (Figure 4) was shown to depend on the different environmental conditions in which the strains were grown, namely on the composition of the medium and the growth temperature. The composition of the EPS produced by SPUR-41 varied across all the different tested conditions. While in SSW at 30 °C, the major components were glucose (32.93% mol) and mannose (58.83% mol), in NSW at 30 °C, there was a relevant decrease in glucose (13.76% mol), and the presence of rhamnose (10.46% mol) was detected. When the growth temperature decreased to 19 °C, arabinose became a component (12.45% mol), and the amount of galactose significantly increased (32.06% mol). Considering SPUR-55, the composition was different from SPUR-41 and with changes across water and temperature variations. In SSW at 30 °C, the EPS was composed mainly of mannose (44.04% mol), glucose (13.86% mol), and galacturonic acid (13.56%mol). Then, when NSW was used for medium preparation, in the assays at 30 °C, the EPS was composed of fucose (5.67% mol), arabinose (19.33% mol), galactose (14.20% mol), glucose (27.10% mol), and mannose (29.86% mol). Moreover at 19 °C with NSW, the EPS also presented a different composition, with the presence of rhamnose (7.63% mol) and galacturonic acid (10.28% mol), while the galactose disappeared (Figure 5). EPS produced by strain SPUR-64 presented some differences when using different cultivation conditions; however, these differences were not as pronounced as in the previous strains (Figure 5). The changes in EPS composition depending on the growth conditions, namely temperature, were already observed for other bacteria, such as *Enterobacter* A47 [54]. This behavior could be related to enzymatic activity since EPS synthesis enzymes are influenced by culture conditions and are a key factor in EPS monosaccharide composition [55].

### 2.4. Bioreactor Cultivation of Strain SPUR-41

Batch cultivation of SPUR-41 was performed in a controlled bioreactor. This strain showed more promising results regarding cellular growth, biopolymer production, and sugar monomer composition of the EPS. Therefore, to obtain further knowledge about growth kinetics and biopolymer production, a batch test was run with NSW and deionized water in the proportion of 75:25 (Figure 6). Further, with the aim of increasing biopolymer production, the initial glucose concentration was increased to 20 g/L.

The exponential phase of SPUR-41 started 2 h after inoculation and took around 8 h, achieving 2.29 g/L of CDW, corresponding to a specific growth rate of 0.25 h^−1^. During the exponential phase, *Brevundimonas huaxiensis* was able to consume 8.38 g/L of glucose. After 8 h, the nitrogen concentration was 0.21 g/L, which probably limited the cellular growth, and for that reason, the culture entered the stationary phase despite the existence of glucose. Nevertheless, in the bioreactor, strain SPUR-41 achieved a higher CDW than in the 200 mL shake flask cultivations. Better control of growth parameters, such as constant pH, higher glucose supplementation, higher availability of oxygen, and higher homogenization of the medium increasing the mass transfer of nutrients and oxygen, contributed to enhanced bacterial growth.

Regarding EPS, production started during the exponential growth phase and continued increasing during the stationary phase, meaning that the EPS production was partly growth associated. Until 15 h, glucose was consumed, and the EPS production achieved was 1.41 g/L; subsequently, the glucose concentration was probably too low, and the bacterium was not able to consume it. However, the maximum EPS concentration was 1.83 g/L, achieved at the end of the assay (35 h), corresponding to a maximum volumetric productivity of 1.30 g/L d^−1^. Between 15 h and 35 h, the increase of 0.42 g/L in EPS was probably due to the consumption of carbon sources present in the yeast extract of the medium. The EPS concentration and productivity achieved were higher than those reported in the literature for other marine-derived EPS producing strains. *Brevundimonas dimituta*, a strain belonging to the same genus of SPUR-41, produced 2 g/L of EPS in 5 days, corresponding to productivity of 0.40 g/L d^−1^ [40]. Other examples include *Halomonas elongate* and *Halomonas halophila*, isolated from Tunisian hypersaline environments, which could secrete 0.16 g/L and 0.11 g/L of EPS, respectively, when grown on complex medium M1 after 120 h of cultivation [21] (Table 1). Nevertheless, some marine-derived bacteria have shown higher performance under similar cultivation conditions, such as *Pseudomonas* sp. MD12-642, which produced 2.5 g/L of EPS in 15 h of cultivation with glucose, corresponding to an EPS productivity of 1.56 g/L d^−1^ [23]. This outcome probably occurred because the used glucose quantity was slightly higher (30 g/L) than that used in this study (Table 1). Furthermore, according to Grobben et al., EPS production is favored by an excess of carbon sources in association with limitations in another nutrient (e.g., nitrogen) [56].

The EPS recovered from the cell-free supernatant indicated that the polymer produced by SPUR-41 was mostly composed of mannose (62.55% mol), glucose (9.19% mol), rhamnose (19.41% mol), and glucuronic acid (4.43% mol). Traces of galactose and galacturonic acid were also identified. The polymer had a high molecular weight (1.2 × 10^5^ Da). These MW values were within the ranges reported for other marine bacteria [57,58].

*Brevundimonas dimituta*, another strain belonging to the same genus of SPUR-41, was reported to be a producer of an EPS that showed 20% antiproliferative activity against myeloid cancer and flocculating activity of 66% [35]. The presence of uronic acids in such high content, especially glucuronic acid, may yield EPS with interesting properties for biotechnological uses, namely for the cosmetics and medical industries. The presence of rhamnose was reported by some authors as having emulsification activity and antioxidant and biofilm-inhibiting properties [26,59,60]. These reports suggest that the EPS produced by SPUR-41 that was cultured in saline medium may be promising for the development of biotechnological products.

## 3. Materials and Methods

Previous to this work, sediment samples were collected in the Estremadura Spur pockmarks field off the coast of Continental Portugal, using a claw Smith McIntyre Grab from depths of 200–350 m, between 31 May and 5 June 2017 [61]. Samples were inoculated using different heat-shock drying methods. Thereafter, the inoculated Petri dishes were incubated at RT (c. 25–28 °C) and monitored periodically over 6 months for bacteria growth (adapted from Roca et al.) [23].

### 3.1. Isolation and Screening of Marine-Derived Bacteria to Produce EPS

Sixty-seven colonies with shiny and slimy morphology (indicative of the strains’ ability to produce viscous EPS), coded SPUR-1 to SPUR-67, were successively transferred onto new media until obtaining pure cultures. The strains were preserved in cryovials with 20% (*v*/*v*) glycerol and stored at −80 °C.

The 67 strains were primarily cultivated in 30 mL shake flasks on medium M1 (adapted from Roca et al. [23]) using synthetic sea water (SSW) (adapted from Akhlaghi Amiri et al.), with the following composition (per L): glucose, 10 g; yeast extract, 4 g; peptone, 2 g; NaCl, 23.38 g; Na_2_SO_4_, 3.41 g; NaHCO_3_, 0.17 g; KCl, 0.75 g; MgCl_2_, 4.24 g; and CaCl_2_, 1.44 g. The pH value of the media was adjusted to 7 with 2 M NaOH before autoclaving [23]. Of the 67 isolated strains, 23 were selected for the screening of their EPS production capacity at a higher scale. During the evaluation of cellular growth and biopolymer production at a 200 mL scale, different media, and temperatures (19 °C and 30 °C) were evaluated for bacterial cultivation, namely media M1 with SSW and M1 medium prepared with natural sea water (NSW) and deionized water in the proportion of 75:25 (*v*/*v*) (per L): glucose, 10 g; yeast extract, 4 g; and peptone, 2 g. The cultures were incubated in an orbital shaker at 30 °C and 200 rpm for 48 h. For all the cultures, samples were collected for determination of the optical density at 600 nm and cell dry weight (CDW) and for EPS quantification.

### 3.2. DNA Extraction, 16S rRNA Gene Amplification, Sequencing, and Taxonomic Identification of Bacteria

All strains were inoculated in 4 mL of medium M1 and incubated at 25 °C for 3 to 7 days with agitation (200 rpm). Total DNA was extracted using the Wizard Genomic DNA Purification Kit (Promega, Madison, WI, USA) as described by the manufacturer with some adjustments [62]. The 16S rRNA gene was amplified using the primers 27F (5′-AGAGTTTGATCCTGGCTCAG-3′) and 1492R (5′-TACGGCTACCTTGTTACGACTT-3′) [63] and NZYTaqII polymerase (NZYtech, Lisbon, Portugal). The PCR products were purified using the NZYGelpure clean-up kit following the manufacturer’s protocols (Nzytech, Lisbon, Portugal). The purified products were quantified and sequenced by the Sanger method using the same primers at STAB VIDA Lda (STAB VIDA, Caparica, Portugal).

### 3.3. Taxonomic Classification and Phylogenetic Analysis

The 16S rRNA sequencing chromatograms were reviewed, and consensus sequences for each forward/reverse pair were created with SeqManPro (Lasergene, DNAstar, Madison, WI, USA). All the sequences obtained were compared to the GenBank rRNA/ITS database by the BLASTn algorithm. The best hits with ≥ 99% sequence identity were aligned using the online version of the muti-sequence aligner MAFFT [64] and the G-INS-i interactive refinement method. A maximum likelihood phylogenetic tree reconstruction of the multi-sequence alignment was performed with the same MAFFT software with 1000 bootstraps. *Thermus aquaticus* (NR_025900.1) was used as an outgroup.

### 3.4. Bioreactor Cultivation

*Brevundimonas huaxiensis* (SPUR-41) cultivation runs were performed in a 3 L bioreactor (Solaris Biotechnology Jupiter Srl), with a 2 L working volume in which sterile conditions were maintained. The medium used in the batch reactor was M1 NSW supplemented with glucose (20 g/L). A 10% (*v*/*v*) inoculum was used. The reactor was operated under controlled conditions of pH 7.0 ± 0.10 by automatic addition of 2 M HCl and 2 M NaOH and a temperature of 30 °C ± 0.5 °C. The dissolved oxygen (DO) inside the bioreactor was kept at greater than 20% by automatically variation of the stirring between 300 rpm and 800 rpm. The air flow rate was maintained at 1 vvm (vessel volume per minute). Foam formation was suppressed by the addition of an antifoam solution (BDH Prolabo—VWR, Radnor, PA, USA). Samples (15 mL) were collected periodically from the bioreactor and centrifuged (8000 rpm, 15 min, 4 °C). The cell-free supernatant was stored at −20 °C for the quantification of glucose, EPS, and nitrogen concentrations, while the cell pellets were used to determine the cell dry weight (CDW).

### 3.5. Analytical Techniques

For determination of the CDW, the cell pellet obtained as described above was washed with deionized water and lyophilized. The CDW was determined gravimetrically by weighing the lyophilized cell pellets. All measurements were performed in triplicate.

Glucose quantification was performed by high-performance liquid chromatography (HPLC) with a VARIAN Metacarb 87H column (BioRad, Heracles, CA, USA) coupled to a refractive index (RI) detector [9]. The analysis was performed at 50 °C using H_2_SO_4_, 0.01 N, as an eluent with a flow rate of 0.6 mL/min. Samples were prepared by diluting the cell-free supernatant in the eluent (H_2_SO_4_, 0.01 N), in 1:20 (first sample of screening growth) and 1:10 (last sample of the screening growth) proportions. All samples were filtered using an Eppendorf membrane filter (0.2 μm). A standard calibration curve was constructed using glucose (99%, Fluka) in a concentration range between 0.01 to 1 g/L.

The cell-free supernatants from the different runs were dialyzed using 12 to 14-kDa molecular weight exclusion membranes (ZelluTrans Carl Roth—Regenerated Cellulose Tubular Membrane, Karlsruhe, Germany) against deionized water at room temperature under constant agitation for 72 h. Sodium azide (10 ppm) was used to avoid biological degradation of the sample. The efficiency of dialysis was controlled by measuring the conductivity (FiveEasy F20, São Paulo, Brazil); when the value was less than 10 μS/m, the dialysis was stopped, and the purified polymer was freeze dried for 48 h (Scanvac, CoolSafe, Lillerød, Denmark). Then, the purified samples were weighed to determine EPS production and kept for further characterization.

For total nitrogen determination, a kit (LCK 388, LATON^®^, Fresno, CA, USA) with a detection range of 20–100 mg/L was used. The test solution (0.2 mL) was placed in a digestion flask; then, the reagents were added as described in the kit, and the flasks were placed on the HT 200S (HACH^®^-LANGE, Ames, IA, USA) digester for 15 min at 100 °C. The flasks were cooled to room temperature. After cooling, the flasks were agitated, and 0.5 mL of the solution was transferred to a new flask, and after 15 min, the absorbance was read on a DR 2800 tm spectrophotometer (HACH^®^).

### 3.6. Biopolymer Characterization

The EPS was characterized in terms of its sugar monomer composition as described by Torres and co-workers [54]. EPS dried samples (approximate 2–3 mg) were dissolved in deionized water (5 mL) and hydrolyzed with 0.1 mL of trifluoroacetic acid 99% (TFA) in a dry bath at 120 °C for 2 h. After cooling to room temperature, 1 mL of the hydrolyzed solution of each sample was filtered using an Eppendorf membrane filter, centrifuged, and finally placed in the specific vial for the analysis. Sugar monomer composition was determined by HPLC using a CarboPac PA10 250 × 4 mm column (Dionex, Sunnyvale, CA, USA) coupled with an AminoTrap 50 × 4 column (Dionex). The analysis was performed at 25 °C with sodium hydroxide (NaOH, 18 mM) as the eluent at a flow rate of 1 mL/min. D-(+)-fucose (98%, Scharlau, Warsaw, Poland), D-(+)-glucose (99%, Fluka, Loures, Portugal), D-(+)-galactose (99%, Fluka), D-(+)-mannose (99% Fluka), D-(+)-galacturonic acid (97%, Fluka), L-rhamnose monohydrate (99%, Fluka), D-(+)-mannose (99% Fluka), D-arabinose (99%, Sigma, St. Louis, MO, USA), and D-glucuronic acid (98%, Alfa Aesan, Haverhill, MA, USA) were used as standards in ranges between 0.5 g/L and 0.005 g/L. The hydrolysate was used for the identification and quantification of the acyl group substituents. The analysis was performed by HPLC, with an IonPac ICE-AS1 9 × 250 mm column (Dionex) coupled to a Photodiode Array PDA ICS series (Dionex) using sulfuric acid (H_2_SO_4_ 0.01 N) as eluent at 30 °C with a flow rate of 0.6 mL min^−1^. The detection was performed at 210 nm. Pyruvate (Alfa Aesar, 98%, Haverhill, MA, USA), succinate (Merck, 99.5%, Rahway, NJ, USA), and acetate (Sigma-Aldrich, 99.8%, St. Louis, MO, USA) solutions were used as standards in concentrations ranging from 1 to 100 ppm.

### 3.7. Molecular Weight Analyses

The analyses were performed on a GPC system from Knauer Smartline equipped with a flow refractive index detector (Knauer, Berlin, Germany). Separation was performed with a Phenomenex Polysep Linear column (300 × 7.8 mm) with a flux of 0.6 mL/min 0.1 M LiNO_3_ and an injection volume of 50 µL. The calibration curve was performed with Pullulan standards in a concentration range of 642 kDa–6.3 kDa (Shodex, Munich, Germany).

## 4. Conclusions

Sixty-seven marine-derived strains, isolated from marine sediments collected in the Estremadura Spur pockmarks field, Portugal, showed the ability to produce EPS. The 23 strains that exhibited the most promising results in the 30 mL assays were tested in 200 mL of medium M1 prepared with SSW. The best results for growth and biopolymer production were achieved using NSW at 30 °C rather than SSW at 19 °C. The bacteria were able to produce between 0.30 and 1.2 g/L of EPS with different sugar compositions, and most were heteropolysaccharides composed of neutral and acidic sugars.

The three strains that presented the best results were *Bacillus* sp. SPUR-55, *Brevundimonas huaxiensis* SPUR-41, and *Bacillus* sp. strain SPUR-64, which yielded EPS production of 1.20, 0.55, and 0.28 g/L, respectively.

SPUR-41 was cultured in a bioreactor using glucose as carbon under non-optimized conditions, achieving 1.83 g/L of EPS, corresponding to volumetric productivity of 1.30 (g/L d^−1^). The EPS synthesized by SPUR-41 was composed of monomers of mannose (62.55% mol), glucose (9.19% mol), rhamnose (19.41% mol), glucuronic acid (4.43% mol), galactose (2.53% mol), and galacturonic acid (1.89% mol). Overall, the achieved results are very promising, encouraging research on marine-derived bacteria to produce value-added products in saline media with potential to be used in a sustainable circular economy approach.

Considering that the microbial biodiversity of marine ecosystems is relatively unexplored, the isolation and identification of new microorganisms that have the potential to provide multiple opportunities for new industrial fields are important. This study allowed for deepening of the research on different marine-derived bacteria for EPS production. For further studies, it would be interesting to fully characterize the structure and biotechnological properties of the EPS produced by these halophilic bacteria and using waste as carbon sources (e.g., waste from the fruit, milk, and beer industries, among others) in combination with saline effluents.

## Figures and Tables

**Figure 1 marinedrugs-21-00419-f001:**
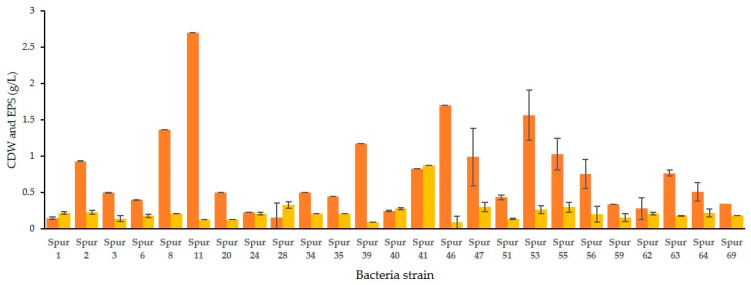
Evaluation of cell dry weight (CDW) (g/L) and exopolysaccharide (EPS) (g/L) production for the selected strains cultivated in M1 medium. Results are the mean of duplicate measurements. (●) CDW g/L; (●) EPS g/L.

**Figure 2 marinedrugs-21-00419-f002:**
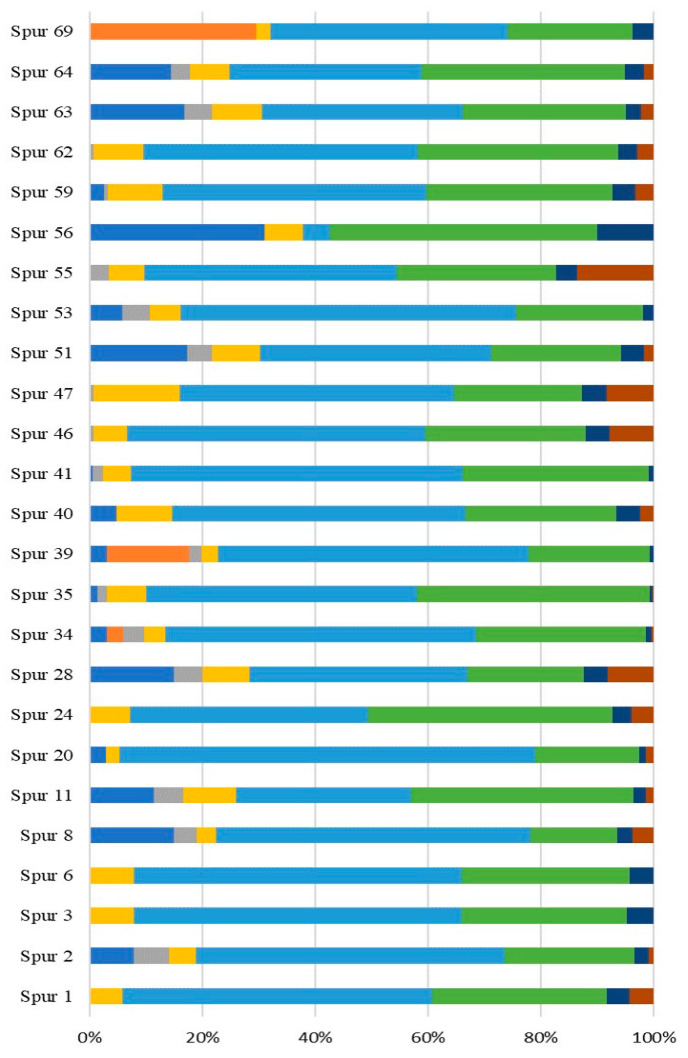
Sugar monomer composition of EPS produced by SPUR strains in 200 mL of M1 SSW. (●) Fucose; (●) rhamnose; (●) arabinose; (●) galactose; (●) mannose; (●) glucose; (●) glucuronic acid; (●) galacturonic acid.

**Figure 4 marinedrugs-21-00419-f004:**
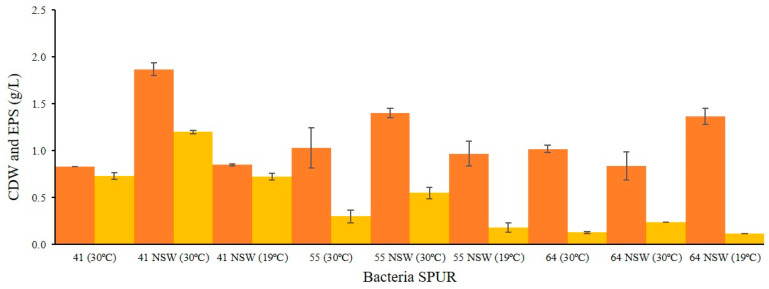
Evaluation of cell dry weight (CDW) (g/L) (●) and exopolysaccharide (EPS) (●) (g/L) production for selected strains cultivated in M1 medium with NSW at 30 °C and 19 °C. Comparison with CDW and EPS from SSW.

**Figure 5 marinedrugs-21-00419-f005:**
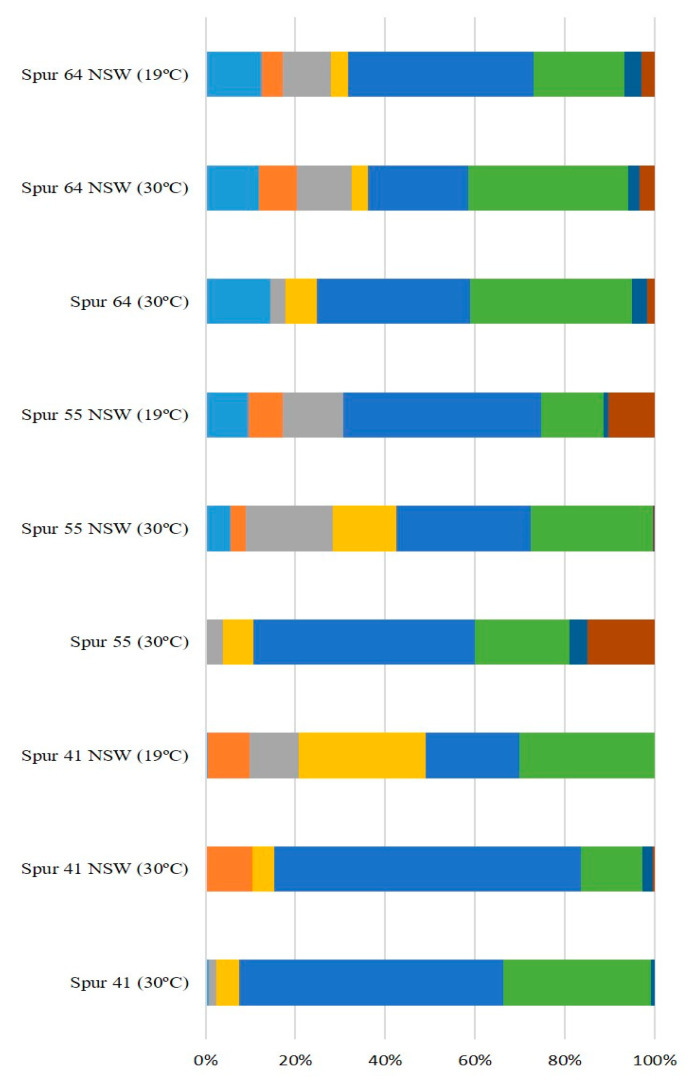
Sugar monomer composition of EPS synthesized by SPUR-41, -55, and -64 in 200 mL of M1 SSW, M1 NSW, and M1 NSW at 19 °C. (●) Fucose; (●) rhamnose; (●) arabinose; (●) galactose; (●) mannose; (●) glucose; (●) glucuronic acid; (●) galacturonic acid.

**Figure 6 marinedrugs-21-00419-f006:**
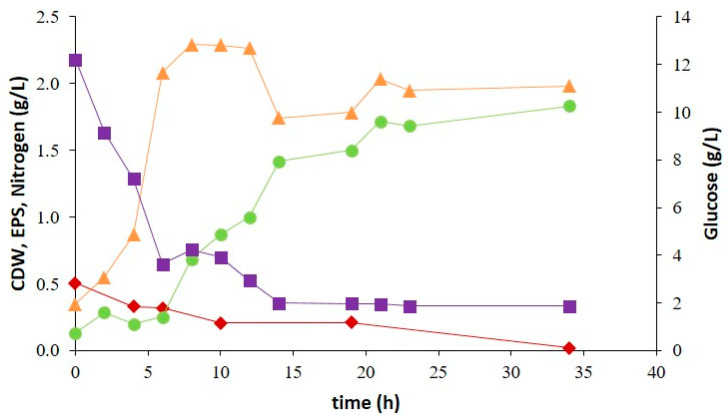
Batch cultivation profile of *Brevundimonas huaxiensis* SPUR-41 on a 2 L bioreactor using glucose as a carbon source, (■) Glucose (g/L) and (♦) nitrogen consumption (g/L), (▲) cellular growth (CDW, g/L), and (●) EPS production (g/L).

**Table 1 marinedrugs-21-00419-t001:** Comparison of EPS production by marine-derived bacteria.

Strain	Location	Carbon Source	Cultivation Time (h)	CDW (g/L)	μ (h^−1^)	EPS (g/L)	r_EPS_ (g/L d^−1^)	Ref.
SPUR-41 Batch	Ocean sediments, Estremadura Spur, Portugal	Glucose	34	2.29	0.25	1.83	1.30	This study
*Brevundimonas diminuta* Batch	Marchica lagoon, Morocco	Glucose	120	0.7	n.a.	2	0.40	[37]
*Pseudomonas* sp. MD12-642 Batch	Ocean sediments, Madeira Archipelago, Portugal	Glucose	15	n.a	0.60	2.5	1.56	[23]
*Halomonas elongate* Batch	Sehline Kerkennah Salt Lake, Tunisia	Glucose	120	n.a	n.a	0.16	0.03	[22]
*Halomonas halophila* Batch	Sehline Kerkennah Salt Lake, Tunisia	Glucose	120	n.a	n.a	0.11	0.04	[22]

## Data Availability

Not applicate.

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
