# Peer review of "Exopolysaccharide Production from Marine-Derived Brevundimonas huaxiensis Obtained from Estremadura Spur Pockmarks Sediments Revealing Potential for Circular Economy"

_marinedrugs, 2023, doi:10.3390/md21070419_

Round 1
Reviewer 1 Report
This paper is mainly on the screening of marine bacteria from Estremadura Spur oceanic sediments as producers of exopolysaccharides, and the study on different media, temperature, and culture volume (30 mL, 200 mL and 3L in bioreactor). This paper shows in particular that the genus Brevundimonas is able to synthetize exopolysaccharides that have very interesting osidic composition.
I’m a little bit surprised by the title of this paper as it mentioned that this screening is toward circular economy. This is not really the aim of your paper. You just mentioned on the lines 492-493 that producing EPS in saline media could help to use the saline effluents. I’m also surprised because I think that industries prefer to avoid have salt in their bioreactors.
I’ve some specific comments to point out:
- Line 76: Hyalurift has never been commercialized. It’s better in your paper to name it EPS HE800
- Line 106: there is no Table 1 available in your manuscript
- Line 119: this is not the Figure 2, but always comments on Figure 1 results
- Lines 126 to 133: In the literature, there are others examples of EPS production from marine bacteria which gave better yields that those you cite. I found your comments are a little bit partial!
- Regarding the Figure 1, it would be more interesting to use different scales for CDW and EPS concentrations
- Lines 158 to 163: There are not so few. I’m sure you’ll find other examples of marine bacteria producing EPS containing fucose (e.g. Zykwinska, A., Marchand, L., Bonnetot, S., Sinquin, C., Colliec-Jouault, S., & Delbarre-Ladrat, C. (2019). Deep-sea hydrothermal vent bacteria as a source of glycosaminoglycan-mimetic exopolysaccharides. Molecules, 24(9), 1703)
- Lines 277 to 284: Do you have some explanations? It would be interesting that you discuss on the osidic composition differences between the two temperatures of growth.
- Legend of Figure 6: This is a lozenge, not a square for nitrogen consumption
- Line 342: No discussion part
- Material and methods part: Methods (M1 and M2) and media used are unclear in the 4.1, 4.2 and 4.3 parts. Could you check if it is correct or not? Could you also rewrite to a better understanding for the reader? May be could you fill a table?
- Line 367: Petri dishes are incubated at RT during 6 months. Is it true?
- Line 417: What is means 1vvm?
- Line 463: What means the “a” “between 0.5 g/L a 0.005 g/L”
- Line 466: please correct H2SO4
Author Response
Reviewer 1
This paper is mainly on the screening of marine bacteria from Estremadura Spur oceanic sediments as producers of exopolysaccharides, and the study on different media, temperature, and culture volume (30 mL, 200 mL and 3L in bioreactor). This paper shows in particular that the genus Brevundimonas is able to synthetize exopolysaccharides that have very interesting osidic composition.
I’m a little bit surprised by the title of this paper as it mentioned that this screening is toward circular economy. This is not really the aim of your paper. You just mentioned on the lines 492-493 that producing EPS in saline media could help to use the saline effluents. I’m also surprised because I think that industries prefer to avoid have salt in their bioreactors.
The title was changed. As these bacterial are able to grow on media with high saline concentrations means that can grow on saline effluents and help to valorise this streams. For industries that produces saline effluents could be an interesting approach.
I’ve some specific comments to point out:
- Line 76: Hyalurift has never been commercialized. It’s better in your paper to name it EPS HE800.
It was corrected on text.
- Line 106: there is no Table 1 available in your manuscript.
The number of tables were corrected.
- Line 119: this is not the Figure 2, but always comments on Figure 1 results
The numeration of the figures was corrected
- Lines 126 to 133: In the literature, there are others examples of EPS production from marine bacteria which gave better yields that those you cite. I found your comments are a little bit partial!
The authors did some changes to the text and add other works reported in literature with EPS production higher than the one achieved in this work. “On the other hand, Roca et al., in similar growth conditions, studied the EPS production by bacteria isolated from Madeira Archipelago ocean sediments achieving concentration higher than the ones obtained in this study (0.99 – 6.88 g/L) [23]. Further, it was reported EPS production between 4.2 and 8 g/L from Egyptian marine bacteria cultivated in 250 mL shake flask with Marine broth and glycerol[24].”
- Regarding the Figure 1, it would be more interesting to use different scales for CDW and EPS concentrations
The figure was changed in order to have gaps between bacteria, however it was kept the same scale for CDW and EPS sincein the authors opinin it is easier to understand the differences between cellular growth and EPS production for each bacteria strain.
- Lines 158 to 163: There are not so few. I’m sure you’ll find other examples of marine bacteria producing EPS containing fucose (e.g. Zykwinska, A., Marchand, L., Bonnetot, S., Sinquin, C., Colliec-Jouault, S., & Delbarre-Ladrat, C. (2019). Deep-sea hydrothermal vent bacteria as a source of glycosaminoglycan-mimetic exopolysaccharides. Molecules, 24(9), 1703)
The sentence was changed to “As other authors reported marine EPS containing fucose in their composition.”
- Lines 277 to 284: Do you have some explanations? It would be interesting that you discuss on the osidic composition differences between the two temperatures of growth.
This differences in monosaccharides composition are probably related with enzymatic activity which are influenced by cultivation conditions. A sentence was added to the manuscript “The changes in EPS composition depending on the growth conditions, namely temperature, was already observed for other bacteria, such as Enterobacter A47 [54]. This behavior could be related with enzymatic activity since EPS synthesis enzymes are influenced by culture conditions and are a key factor on EPS monosaccharide composition [55].
- Legend of Figure 6: This is a lozenge, not a square for nitrogen consumption
The symbol was corrected for a lozenge.
- Line 342: No discussion part
The discussion part was removed, since in this work results and discussion are presented together.
- Material and methods part: Methods (M1 and M2) and media used are unclear in the 4.1, 4.2 and 4.3 parts. Could you check if it is correct or not? Could you also rewrite to a better understanding for the reader? May be could you fill a table?
The numeration was re-written and the section 3.1 and 3.2 was removed from the manuscript since it is the description of the sediments collection and sediment sample inoculation, work that is not presented in this manuscript.
- Line 367: Petri dishes are incubated at RT during 6 months. Is it true?
Yes.
- Line 417: What is means 1vvm?
Vessel volume per minute = air flow rate/operating volume
- Line 463: What means the “a” “between 0.5 g/L a 0.005 g/L”
The "a" was substituted by "to".
- Line 466: please correct H2SO4
It was corrected.
Reviewer 2 Report
1. Abstract: line 15: EPS are not a new biomolecules
2. Brevundimonas huaxiensis – please use an italic
3. Lines 20-24 – please rewrite, avoiding details about breeding conditions and yield (all in results)
Mannose, glucose, rhamnose etc. are a monomers so it is not necessary use this world
Line 25: “small content of galactose”??? what about galacturonic acid content?
Please put short information which exactly acyl groups are in the structure.
4. Please use once entered shortcuts for example: Exopolysaccharide, EPS
5. Line 72: seven different sugar residues
6. Line 100: innovative EPS???
7. Line 101: monosaccharide composition
8. Where the Table 1 is?
9. Line 105: biosynthesize
10. What does mean: “best producers”
11. There are 25 strains described on the Figure 1
12. Line 116: please put numbers with two places after decimal\
13. Line 122: if the pH has not been controlled during the runs – why the Authors know the 6-7 values? This sentence is only a conjecture.
14. Line 126: please remove “very”
15. EPS is a polymer which is build by sugar residues
16. Rhamnose is exactly deoxy-hexose
17. Please use sugar residue instead of sugar
18. Line 148: biofilm
19. Line 155: film-forming properties????
20. Line 183, 194: please start use: B. hauxiensis, B. aureus
21. Line 196: ….Manp…..Glcp…
22. Please standardize the temperature description
23. Line 330: mass unit needed
24. Line 342: Please remove all paragraph – the discussion has been made in the text above
25. Line 475: 642KDa-6.3 KDa?
26. Funding and Acknowledgements are the same
27. Please prepare references according to the instructions
Author Response
- Abstract: line 15: EPS are not a new biomolecules
The word new was removed
- Brevundimonas huaxiensis – please use an italic
It was corrected.
- Lines 20-24 – please rewrite, avoiding details about breeding conditions and yield (all in results)The text was changed.
Mannose, glucose, rhamnose etc. are a monomers so it is not necessary use this world
The word monomers was removed.
Line 25: “small content of galactose”??? what about galacturonic acid content?
"small content" was removed from the sentence.
Please put short information which exactly acyl groups are in the structure.
It was corrected.
- Please use once entered shortcuts for example: Exopolysaccharide, EPS
The comment was taken into consideration.
4. Line 72: seven different sugar residues
It was corrected
- Line 100: innovative EPS???
The word innovative was substituted by the word rare.
- Line 101: monosaccharide composition
It was corrected.
7. Where the Table 1 is?
The reference at Table 1 in the text was deleted.
- Line 105: biosynthesize
The word was changed.
- What does mean: “best producers”
It was changed to higher yielded producers.
- There are 25 strains described on the Figure 1.
These 25 five strains are the one that presented the higher EPS production. The number was corrected on the text.
- Line 116: please put numbers with two places after decimal\
It was corrected.
- Line 122: if the pH has not been controlled during the runs – why the Authors know the 6-7 values? This sentence is only a conjecture.
The authors know the values because the pH was measured at the end of the runs.
- Line 126: please remove “very”
It was removed.
- EPS is a polymer which is build by sugar residues
- Rhamnose is exactly deoxy-hexose
- Please use sugar residue instead of sugar
- Line 148: biofilm
- Line 155: film-forming properties????
- Line 183, 194: please start use: B. hauxiensis, B. aureus
- Line 196: ….Manp…..Glcp…
- Please standardize the temperature description
- Line 330: mass unit needed
- Line 342: Please remove all paragraph – the discussion has been made in the text above
- Line 475: 642KDa-6.3 KDa?
The comments above, from 15 to 25, were taken into consideration and corrected.
- Funding and Acknowledgements are the same
The Funding and Acknowledgements sections were changed.
- Please prepare references according to the instructions
References were revised.
Reviewer 3 Report
The manuscript presents a screening process involving 67 marine-derived bacteria to assess their ability to produce EPS. The evaluation was conducted using shake flasks and then a bioreactor, with a focus on the strain Brevundimonas huaxiensis SPUR-41. The manuscript is well-written and, with some minor revisions, can be accepted for publication:
General comments:
Please maintain consistency in formatting. For example, if you write "37℃" do not include a space like "37 ℃". Regarding (%), it should always be without space between the numerical value and the symbol , for example (9.19% mol).
Abstract:
Line 19: The name of strain should be Italics.
Discussion:
Line 342: I don’t see any discussion.
Materials and methods:
Section 4.6, Line 417: Please write what is vvm and also write it in Italics.
Section 4.7, Line 427:
-Please specify the manufacturer.
- After reading the reference (8), I discovered that they utilized a refractometer, which functions similarly to a Refractive Index (RI) detector. Kindly verify whether you employed an Infrared detector (IR) or if it was a typographical error.
Section 4.8:
Line 457: Please add the unit to (250 × 4), also to line 458. Please use symbol × instead of x.
Line 466: Please use superscript for 2 and 4 in H2SO4.
Conclusion:
Line 484 and 485: Please write the strains in italics.
Author Response
The manuscript presents a screening process involving 67 marine-derived bacteria to assess their ability to produce EPS. The evaluation was conducted using shake flasks and then a bioreactor, with a focus on the strain Brevundimonas huaxiensis SPUR-41. The manuscript is well-written and, with some minor revisions, can be accepted for publication:
General comments:
Please maintain consistency in formatting. For example, if you write "37℃" do not include a space like "37 ℃". Regarding (%), it should always be without space between the numerical value and the symbol , for example (9.19% mol).
It was corrected.
Abstract:
Line 19: The name of strain should be Italics.
It was corrected.
Discussion:
Line 342: I don’t see any discussion.
It was removed. The discussion was presented together with results-
Materials and methods:
Section 4.6, Line 417: Please write what is vvm and also write it in Italics.
Section 4.7, Line 427:
-Please specify the manufacturer.
- After reading the reference (8), I discovered that they utilized a refractometer, which functions similarly to a Refractive Index (RI) detector. Kindly verify whether you employed an Infrared detector (IR) or if it was a typographical error.
It was corrected to refractive index (RI) detector.
Section 4.8:
Line 457: Please add the unit to (250 × 4), also to line 458. Please use symbol × instead of x.
Line 466: Please use superscript for 2 and 4 in H2SO4.
It was corrected.
Conclusion:
Line 484 and 485: Please write the strains in italics.
It was corrected.